# Perceived Health Impacts of Watershed Development Projects in Southern India: A Qualitative Study

**DOI:** 10.3390/ijerph17103448

**Published:** 2020-05-15

**Authors:** Adithya Pradyumna, Arima Mishra, Jürg Utzinger, Mirko S. Winkler

**Affiliations:** 1Swiss Tropical and Public Health Institute, P.O. Box, CH-4002 Basel, Switzerland; juerg.utzinger@swisstph.ch (J.U.); mirko.winkler@swisstph.ch (M.S.W.); 2University of Basel, P.O. Box, CH-4003 Basel, Switzerland; 3Azim Premji University, PES Campus, Hosur Road, Bengaluru 560100, India; arima.mishra@apu.edu.in

**Keywords:** agriculture, India, livestock, nutrition, vector-borne diseases, watershed development project

## Abstract

Watershed development (WSD) projects—planned for over 100 million ha in semi-arid areas of India—should enhance soil and water conservation, agricultural productivity and local livelihood, and contribute to better nutrition and health. Yet, little is known about the health impacts of WSD projects, especially on nutrition, vector breeding, water quality and the distribution of impacts. We conducted a qualitative study to deepen the understanding on perceived health impacts of completed WSD projects in four villages of Kolar district, India. Field data collection comprised: (i) focus group discussions with local women (*n* = 2); (ii) interviews (*n* = 40; purposive sampling) with farmers and labourers, project employees and health workers; and (iii) transect walks. Our main findings were impacts perceived on nutrition (e.g., food security through better crop survival, higher milk consumption from livestock, alongside increased pesticide exposure with expanded agriculture), potential for mosquito larval breeding (e.g., more breeding sites) and through opportunistic activities (e.g., reduced mental stress due to improved water access). Impacts perceived varied between participant categories (e.g., better nutrition in woman-headed households from livelihood support). Some of these findings, e.g., potential negative health implications, have previously not been reported. Our observations informed a health impact assessment of a planned WSD project, and may encourage implementing agencies to incorporate health considerations to enhance positive and mitigate negative health impacts in future WSD projects.

## 1. Introduction

India is an agrarian nation. Indeed, data for the year 2011 revealed that among the total working adult population of 482 million, 263 million (54.6%) were dependent on agriculture for their livelihood, including 144 million (30.0%) as wage labourers [1]. Of the net-cultivated area, around 56% is rain-fed [2]. Agriculture in India faces several challenges, such as land degradation [3,4] and depletion of groundwater [5]. India is also vulnerable to climate change [6,7], which is projected to affect agriculture through increased mean temperature, increased mean precipitation and a decrease in rainy days [8]. These challenges are of particular concern to marginalised and small farmers [9] (those owning less than one and two ha of land, respectively [1]). In order to address these challenges in semi-arid areas, several pathways have been suggested, including watershed development (WSD) [10].

The Government of India has defined WSD as “*the conservation, regeneration and the judicious use of all the resources—natural (land, water, plants, and animals) and human—within the watershed area. Watershed management tries to bring about the best possible balance in the environment between natural resources on the one side and man and animals on the other*” [11]. WSD projects involve several physical, ecological and social interventions. Physical interventions support soil and water conservation through structures such as contour bunds, gully plugs, plantation and vegetative checks, ponds and wells, trenches, check dams and land-levelling [12]. Ecological interventions include support to horticulture and pastures [13]. Social interventions include the creation of local management institutions and livelihood opportunities [13,14]. These interventions, which are made at the level of a cluster of villages in semi-arid areas over a period of 3–5 years, are expected to enhance and sustain agriculture and livelihoods [13]. While soil and water conservation projects have been implemented by the Government of India and non-governmental organisations (NGOs) through various schemes over decades, WSD projects are in operation since the early 1990s, and common guidelines were reissued in 2008 (and revised in 2011), which have since formed the basis for WSD projects [13].

Literature from high-income countries have addressed “watersheds” as a geographical unit that needs to be managed appropriately to have optimum impacts on ecology, human development and public health [15,16,17]. Water quality in rivers and surface water bodies was the main health issue addressed [18,19,20]. A review on watershed management and public health highlighted specific concerns such as microbiological contaminants, and indicated that health and social concerns have, thus far, not been reported adequately [15]. Prior research from Ethiopia on the impact of soil and water conservation in semi-arid areas revealed that most farmers perceived positive impact of water-holding structures on agricultural productivity [21].

A meta-analysis of over 600 WSD project evaluations in India showed positive impacts on increasing income, improving local employment opportunities (151 person-days per ha), higher crop yields, greater cropping intensity (35.5%), groundwater recharge, decrease in run-off (45%) and soil loss (1.1 ton per ha), contributing to social capital and reduction in poverty [22]. Positive impacts on livelihoods and milk production were also reported by an NGO in the state of Maharashtra in western India [23,24], where growth monitoring and child nutrition were incorporated as part of WSD project activities [20]. A participatory evaluation from northern India also revealed perceived positive impacts on income, awareness and women’s empowerment [25]. These studies illustrate the impact of WSD projects on well-being and the local environment.

In addition to potential socioeconomic impacts, WSD projects have considerable potential for community health impact through influencing agriculture, water and livelihoods. From the Indian context, only three peer-reviewed journal articles describing the linkages between WSD and health were found, all conducted by Nerkar and colleagues. From a cross-sectional study, they found lower diarrhoea prevalence in watershed villages as compared to comparison villages [26]. This quantitative paper was complemented by two qualitative studies. The first reported local people having perceived reduced water-borne diseases, reduction on workload on women due to water access, reduction in alcohol consumption, improved socioeconomic status (SES), support to education and women’s empowerment [27]. The second qualitative study with local health and development workforce reported perceived reduction in water scarcity, better access to sanitation and improved local agriculture, resulting in positive impacts on SES, educational and health status in communities affected by the WSD project [28]. A limitation of the studies by Nerkar and colleagues is that they were conducted in a hilly tribal area in Maharashtra, thus representing one specific geographical and socio-cultural context. WSD projects in other regions may include additional locally-relevant activities potentially leading to different types of impacts [29]. In addition, other health impact pathways, for instance, on nutrition, vector breeding, water quality and also the distribution of impacts within the population were not adequately explored. In summary, considerable knowledge gaps exist on the interlinkages between WSD projects and associated health impacts in India.

The objective of this paper was to describe and better understand the types of WSD project-related health impacts (positive and negative) as perceived among community members and local organisations from a semi-arid agricultural area in Kolar district, southern India. The main value of this pursuit is due to WSD projects being planned for degraded and rain-fed regions in India covering an area of 146 million ha and 85 million ha, respectively, poised to benefit millions of households [13]. A high-level committee also envisioned WSD projects as an “*umbrella to unite all development programmes*” in rural areas, including those instituted by the Ministry for Health and Family Welfare on issues such as undernutrition [30]. Therefore, this study contributes to the literature on how completed WSD projects have impacted health, by addressing some of the identified gaps. The research was conceptualised as an exploratory study to be followed by a health impact assessment (HIA) [31] of a newly planned WSD project in the same region.

## 2. Materials and Methods

### 2.1. Design and Study Setting

We designed a qualitative study, incorporating various methods, including semi-structured interviews, focus group discussions (FGDs) and transect walks. Interviews and FGDs were conducted with local people, NGO officials, field workers (project employees), local health workers, an environmental health academician and a WSD expert. The theoretical underpinning of our study was that WSD projects affect local well-being, the environment and livelihood—which are all determinants of health—and therefore also have potential to impact health, which can be appreciated and reported by people who participated. In addition, though our approach was largely interpretative (through asking open-ended questions on perceived health impacts), we also incorporated positivist elements (specifically prompting participants to consider plausibly relevant diseases informed by existing literature, as discussed later), to address our study objectives. These theoretical aspects informing the research design were described based on the qualitative research manual by Green and Thorogood [32].

Our study was carried out in Kolar district in Karnataka state (Figure 1), classified under the “South Eastern Dry Zone”, where a high vulnerability has been projected for surface and groundwater, with increased winter-time drought occurrence [33]. Purposive sampling was applied for the selection of study villages. In a first step, NGO officials were invited to list and describe all recently completed WSD projects (four projects in total, completed 2–10 years prior to our study, as illustrated in Figure 1). Subsequently, one village from each WSD project was selected (Table 1).

### 2.2. Data Collection

Initially, two FGDs adhering to methods described by van Eeuwijk and Angehrm [35] were conducted with women of working age in the first study village. Following this, a total of 40 semi-structured interviews employing methods described by Britten [36] were conducted with the following types of participants: (i) local people (both land-owning and landless) (*n* = 26); (ii) local health workers (*n* = 5); (iii) field project staff (*n* = 4); (iv) project managerial staff with experience in WSD (*n* = 3); (v) one senior environmental health academician; and (vi) one senior WSD project manager from another region and organisation. The sampling strategy was “purposive” [32,37] to ensure representation based on land ownership, access to irrigation, gender and caste. Participation in WSD project activities was a pre-requisite for those selected for interviews as we were interested in understanding perceived impacts, which would be better articulated by those who participated [32]. As specific activities of the WSD project were targeted to sub-groups of the population, we interviewed land-owning persons (for insights on impacts of agriculture-related activities), landless persons (for insights on impacts of livelihood activities), men (for insights on activities performed by men) and women (for insights on impacts on activities performed by women) accordingly. The number of participants for interviews was not pre-decided. Interviews were continued until no additional insights on health impacts of WSD projects were obtained from further interviews, and we were able to validate findings across study sites and participant categories [32]. Most of the interviews were conducted in July and August 2018. A few additional and follow-up interviews were carried out in December 2018 and November 2019. The time interval between phases of interviews were used to analyse the data which helped frame new issues to explore in future interviews. These intervals did not affect the nature of the responses, as all the projects had already been completed 2–10 years prior to our data collection.

Participants were introduced to the interviewer by a local liaison person. The interviews were conducted by the first author in Kannada language (except those with managers, experts and academicians were conducted in English). Potential participants were first informed about the research. After obtained informed consent, the interviews were conducted and audio-recorded. Most interviews with land-owning farmers were done on the spot in the agricultural fields. Interviews with the landless were conducted at their homes or in a community hall. FGDs were conducted at a vacant house. Occasionally, children and other family members were present during an interview. A field research journal was maintained to make notes of observations and interactions based on the method described by Phillippi and Lauderdale [38].

Following initial interviews in each village, a transect walk, as described by Loewenson and colleagues [39], was conducted with the local liaison to see the WSD-related structures mentioned in the interviews and to observe other factors that might influence health. Interview guides were used for the interviews and FGDs. The topics included in these guides were based on impact pathways described in relevant literature [23,24,26,27,28,40], brainstorming on potential health impacts and impact pathways (Figure 2) and insights gained from initial interviews [32]. Broadly, the topics covered in the interviews were: status of agriculture (i.e., productivity, crop choice and irrigation), wage labour, access to food, water, sanitation and health (i.e., infectious diseases, non-communicable diseases, injuries and access to healthcare), participation in WSD projects and perceived impacts of WSD projects on agriculture (i.e., productivity, output, irrigation and crop choices), water (i.e., conservation and access), livelihood, food (i.e., food security and dietary diversity) and health (e.g., nutrition, vector-borne diseases (VBDs), pesticide use, injuries and access to healthcare).

### 2.3. Data Management and Analysis

The recorded interviews were transcribed directly into English by the first author (AP). Some initial transcripts were reviewed by the second author (AM). The analysis used the framework method described by Gale and colleagues [41], with additional inputs on thematic analysis from Braun and Clarke [42]. The transcripts were read and re-read, and a coding scheme was developed through deductive and inductive approaches. The transcripts were loaded onto OpenCode 4.03 software created at Umeå University (Umeå, Sweden) [43]. After coding a few transcripts, a working analytical framework was developed, and the codes and categories were defined. The framework was applied to the entire data set and updated when necessary. Coding was done by the first author. The codes are listed in Table 2. After coding, the data were summarised and charted into the framework matrix. Interpretation of the data was done by characterising the data and identifying agreement between data especially within groups in a village, between projects and between local people and key informants. Attempts were made to understand the reasons for the emergence of the identified phenomena [41]. To describe and explain perceived health impacts of WSD projects, themes were identified through further interrogation within and between categories [41,42]. The consolidated criteria for reporting qualitative research (COREQ) were used to adhere to reporting standards [44].

The first author (AP, male) is a medical and public health graduate, currently pursuing a PhD in epidemiology. He has worked on climate change vulnerability in WSD project areas for five years. He is trained in qualitative research. AM (female) is a medical anthropologist (with a PhD), currently a university professor in India. She is familiar with local nutritional and development challenges. JU (male) is a university professor (with a PhD) with vast experience in epidemiology, environmental sciences and public health, and has supervised PhD students from several countries. MSW (male) is an environmental epidemiologist (with a PhD), head of the HIA research group at the Swiss Tropical and Public Health Institute, and has worked in several countries on health impacts of developmental projects.

### 2.4. Ethical Considerations

This study obtained ethical approval from the Padmashree Institute of Clinical Research in Bengaluru, India (reference no. IEC-BIO-004; approval date: 10 August 2018) and the Ethics Commission of Northwestern and Central Switzerland (EKNZ, reference no. BASEC Nr Req-2018-00839, approval date: 19 October 2018). The purpose and procedures of the study were explained to the participants and informed consent (in the local language) was obtained prior to the interviews. An information sheet prepared in local language about the project and the contact details of the main investigator was handed over to participants and local liaisons. The data have been stored on a server at the Swiss Tropical and Public Health Institute (Basel, Switzerland), in an anonymised manner. At the time of writing this manuscript (April 2020), dissemination of findings to the NGO has already been initiated.

## 3. Results

The themes ‘Pathways towards nutritional impacts’, ‘Impacts on disease vector ecology’ and ‘Health impacts of opportunistic activities’ addressed the emergent health-related impacts of WSD projects in the study area. Details of the participants are provided in Table 3.

Quotes by local people have been labelled with gender, land ownership, caste (scheduled caste as SC; scheduled tribe as ST; other castes as GC) and age. Quotes by key informants are labelled with type of informant and a serial number.

### 3.1. Pathways towards Nutritional Impacts

#### 3.1.1. Direct Impacts on Food Security and Nutrition

Pathways for direct impact on food security and nutrition included better survival of staple crops, fruit trees, vegetable production and milk from livestock. It was perceived that rain-fed staple crops such as finger millet were “*better-off*” (land-owning female, GC, 38 years) as WSD structures helped them survive “*even when there is no rain for some days*” (land-owning female, GC, 60 years), leading to “*increased*” production (land-owning male, GC, 38 years). Additionally, through improved soil and moisture, “*land for agriculture has increased*” (land-owning male, GC, 63 years) as fallow lands have been “*brought under production*” (NGO manager 2). Both these aspects were especially important for small farmers without irrigation access.

Simultaneously, during recently implemented WSD projects, farmers were encouraged to adopt water-conserving technologies such as drip-irrigation using available subsidies. Mulching paper was also privately adopted by vegetable farmers. These technologies helped increase yields as they doubled the irrigated area “*with the same volume of water*” (land-owning male, GC, 36 years). While these technologies and other interventions such as farm ponds led to more farmers taking up vegetable cultivation, they did not neglect the cultivation of staple crops for their household needs:
“For home use, we grow finger millet surely, and pigeon peas and flat beans. If we ignore that, and plant other crops in those lands and don’t get returns…it will be difficult.”(land-owning male, GC, 35 years)

However, many small farmers removed the WSD structures from their fields to reclaim land that was “*lost to the bunds and trenches*” (land-owning male, GC, 63 years). In two areas, enrolment itself was low due to the feeling that land “*will get wasted*” (land-owning male, GC, 38 years). This may have affected food security of their households. Unrelated to WSD, in some areas the recent “*nuisance*” of deer, pigs, elephants and peacocks had affected production of staple and commercial crops (land-owning male, GC, 35 years).

WSD may have marginally improved vegetable production in two ways: increased land productivity, and supporting drip irrigation and farm ponds. These impacts were evident in one remote village where vegetable production reportedly increased considerably after the project. Farmers also mentioned that cultivated vegetables were consumed regularly in their households, and wage labourers received “*beans, tomato, radish, carrot, beetroot, whatever was being grown*” (landless female, ST, 45 years), especially “*smaller and broken*” vegetables, to take back home (landless female, ST, 55 years). Vegetables also reportedly became more easily available at the villages.

Saplings of guava, pomegranate, custard apple, mango and plum were provided for planting on private and common lands through the projects, but “*few of them survived*” due to poor rains (landless female, ST, 45 years). Better survival and growth was noticed for “*trees near the farm ponds and check dams*” (NGO fieldworker 4), the former often adopted by those with larger holdings. In more recently completed WSD projects, the trees were still young, and people mentioned that they would “*eat the fruit*” when they were borne (local health-worker 3). Landless families that invested the financial aid in livestock were now able to “*consume milk regularly*” (landless female, ST, 37 years). This was perceived to have contributed to children becoming “*healthy*” (NGO manager 3).

On the other hand, it was felt that people were “*using lots of pesticide*” in vegetable farms (NGO manager 3), contributing to increased exposure to pesticides. However, improved soil quality has reportedly allowed farmers to manage “*with less fertiliser*” (land-owning male, GC, 63 years). Another impact was “*increased workload*” on women in households cultivating vegetables through irrigation (NGO fieldworker 1). Women were involved in “*sowing seeds and removing weeds*” (NGO fieldworker 1), which need to be done manually, unlike men’s work that “*can be done by machines*” (land-owning female, GC, 35 years).

#### 3.1.2. Income-Mediated Impacts on Food Security and Nutrition

Several landless households in remote villages did “*not have enough to eat*” but the situation became “*much better*” following the WSD project (landless female, ST, 45 years). These households were also often woman-headed and from SC or ST. While the governmental “*ration shop*” contributed to food security (FGD, landless women), livelihood support through WSD projects considerably enhanced food security and nutrition through the income route. NGO officials perceived that “*more varieties*” of food products were consumed following the projects due to better income and awareness (NGO manager 2). This was reciprocated, especially by beneficiaries of livelihood support who felt “*everything has changed*” (landless male, ST, 28 years), including increased consumption of milk and meat.

Support for livelihood was directed to landless households and were carried out through creation of new and strengthening of existing self-help groups (SHGs, savings groups composed of around 10 members). The mechanism was through provision of small grants and loans, for example in one project it amounted to “*Rs. 10,000 as grant and Rs. 15,000 as loan*” (NGO manager 3). Women also were trained on “*taking loans, how it could be used for livelihoods and investment*” (landless female, ST, 45 years). The “*focus on animal husbandry has been high*”, which provided “*good returns*” (NGO fieldworker 4). “*People who were unable to go for daily wage*” (land-owning female, GC, 60 years), were now able to earn a reasonable livelihood by “*taking care of cows and sheep*” (landless female, ST, 37 years).
“I had bought one (cow), and eventually bought four. Now I have two, but have kept 20 sheep too. I didn’t have cows or sheep earlier. I only worked for daily wage.”(landless female, ST, 37 years)

Similarly, some poor households that invested in livelihoods were able to build on it to improve their quality of life considerably. One beneficiary who earlier was a wage carpentry labourer “*bought tools and started own business*” using grants and loans provided (landless male, ST, 28 years). Eventually he “*hired people to work*” for him and also “*bought two bikes*” (landless male, ST, 28 years).

One participant “*had planted 400 trees of many varieties*” through the project and felt he “*would have been rich*” had it rained at the right time (land-owning male, GC, 50 years). A prolonged drought shortly after the completion of the WSD project which further contributed to agricultural debt and unpredictability of labour opportunities, may have impacted on potential benefits of the WSD project. Indeed, several households started travelling regularly for work due to prospects of “*better income in Bengaluru*” and elsewhere (land-owning male, GC, 50 years). Members of some households reportedly travelled during the dry season, while some have migrated permanently. For some of these households, it may have been because of inability to invest the livelihood grant.
“They gave us Rs. 7,000…told us to buy a goat and grow the money...but we didn’t …we just used it for our regular expenses.”(landless female, SC, 68 years)

In another village, one woman cared for 20 custard-apple trees planted on common lands during a WSD project. She “*sold the fruit*” regularly for side-income (landless female, ST, 55 years). Large farmers of perennial rain-fed crops such as mango were highly benefited by WSD projects, but these farmers were already earning well, and less concerned about food security.
“We put bunds in 10 acres of mango plantation. I was not getting good yield earlier. Once it rained next year, the water was held in the soil in my field and the plants developed well. We got 5 tonnes earlier, then we got 10–15 tonnes and now we get even up to 25 tonnes.”(land-owning male, GC, 63 years)

### 3.2. Impacts on Disease Vector Ecology

As part of the WSD projects, water-holding structures such as check dams, farm ponds and drinking-water troughs for cattle were constructed and open wells were encouraged (Figure 3). In addition, lakes and ponds were revitalised. These structures could potentially breed mosquitoes if they hold water for sufficient time. However, local people and NGO officials did not notice any increased mosquito nuisance or larval breeding in these structures and did not perceive any risk.

WSD projects “*encouraged conserving water using open farm ponds*” of which several were created (NGO manager 3). These could hold water “*between rainy season and summer for 6 months*” (local health-worker 2), but those ponds with loose soil did not hold water “*for even 8 days*” after rainfall (NGO fieldworker 4). Those with large fields empty the farm pond each day for irrigation, and then refill it with groundwater.

Some “*mini-check dams were constructed*” (NGO manager 3), which held water “*for 2–3 months*” following rains (NGO manager 3), and one was noted to “*always have at least some water*” (land-owning male, ST, 32 years). In one project area, public “*water troughs for cows*” were made that reportedly were always full (land-owning male, GC, 36 years). “*Open wells were promoted*” too (NGO manager 2); however, most “*wells don’t hold rainwater*” for long (land-owning male, GC, 63 years). One well near a mini-check dam held “*water for 6 months*” following adequate rains (land-owning male, GC, 38 years). One lake re-vitalised by creating a mini-check dam through the project “*gets filled*” after rains (NGO fieldworker 4).

It was perceived by local people and NGO workers that these water-holding structures had not contributed to any increase of mosquito nuisance. One official felt that “*mosquitoes usually go where there is pollution, like in the drains*” and had “*never seen mosquitoes*” in water-holding structures (NGO manager 2). One fieldworker opined that “*only if there is less water some mosquitoes may grow*” (NGO fieldworker 3). Local people mentioned that check-dams and farm ponds “*have been made outside the village. So there is no problem about mosquitoes*” because of the distance (land-owning female, GC, 35 years). Most local people also perceived mosquitoes to be mainly associated with stagnant water in “*ditches which were made next to our houses*” (FGD, women of SHG) and forested areas with “*lots of plants*” (land-owning male, GC, 38 years).

One environmental health academician commented that “*check dams can breed Anopheles*”, wells are “*a major source of Anopheles*” and troughs holding some water “*will support breeding of Aedes*” (health academician 1). Structures such as compost pits, which were created in some projects “*can be huge Culex-breeding sites*” (health academician 1). It was also discussed that *Anopheles* species could travel “*up to 2 km*” (health academician 1). Hence, structures located just outside the village were not far enough to be protective. The academician also confirmed that people get bitten in forested areas because *Culex* mosquitoes reside there, as they favour stagnant water in open drains for breeding. Bio-environmental control using *Poecilia* (guppy) and *Gambusia* fish species, already being practiced for hot-spots by local health services, was recommended for check-dams, wells and ponds, alongside other fish for nutrition:
“They can use carpio, which is edible and can grow to 600–700 g and is a protein source, and use gambusia along with that.”(Health academician 1)

However, fish rearing was not practiced because reportedly “*people keep taking the fish*” (land-owning male, ST, 32 years) and also because it was felt that fish “*don’t survive well in tarpaulin-lined ponds*” (land-owning male, GC, 63 years). Local health workers mentioned visiting households regularly to inspect water containers and “*inform people to clean them*” (local health-worker 2). Fish release had reportedly been done once in some project villages a few years ago by the health department, but local people had “*never heard about this*” (NGO fieldworker 1). This is likely because fish inoculation was only done in public water bodies, and there have not recently been cases of malaria reported locally.

### 3.3. Health Impacts of Opportunistic Activities during WSD Projects

Various additional health-specific or health-determining activities were conducted. These were either entry-point activities, awareness sessions or support to leverage welfare schemes. Some health-related activities were conducted to build trust with the local community, such as “*cancer detection, eye and dental camps*” (NGO manager 3). “*Veterinary camps*” were also conducted to support livestock health (land-owning male, GC, 36 years). Local people were also encouraged and supported to enrol for “*health insurance schemes such as Rashtriya Swasthya Bima Yojana*” (NGO manager 3), but most respondents “*didn’t follow up*” (landless female, SC, 40 years) to enrol. Assistance to sign up for health and life insurance were, like SHGs, also supported outside of WSD projects. Some poor and woman-headed households were “*managing mainly because of the self-help group*” (landless female, SC, 40 years), including high expenditures accessing healthcare.

One training on “*menstrual hygiene*” was conducted for SHG members and adolescent girls (NGO manager 3). “*Awareness about hygiene and toilets was also given as there are grants from panchayat*” (NGO manager 2). One older WSD project had also provided financial “*support for building toilets*” (land-owning male, GC, 50 years). In addition, voluntary service was organised for “*cleaning up the crèche and school*” (NGO fieldworker 4), “*local water bodies and planting trees near borewells*” (NGO manager 3). Women from remote villages especially appreciated the field trip outside the district where they learned about “*agriculture, plants, fodder and about the villages*” (landless female, ST, 55 years). Farmers remembered training sessions about “*conserving water, about ground water conservation, the need for check dams and avoiding use of groundwater to conserve for our children*” (land-owning male, GC, 63 years).

Respondents in a remote tribal hamlet appreciated the entry-point activities, as their pressing needs for a water tank and community hall were addressed. Earlier they had to “*go down the hill to collect water*” (land-owning female, ST, 65 years), and now they were “*sure to get water after work in the evening*” (landless male, ST, 38 years), which contributed to physical and “*mental relief*” (NGO manager 2). Villages in other project areas already had water supply through governmental initiatives. Diarrhoeal diseases were “*not seen*” commonly in these villages (local health-worker 2).

Project field workers also helped local people access welfare schemes provided by NGOs such as subsidised cooking-gas connection. This was opined to have prevented the forests from “*getting damaged*” (NGO fieldworker 4) and would have also reduced exposure to indoor air pollution. This NGO-based subsidy preceded the governmental scheme by a few years.
“We bought it (gas connection) through the SHG itself. Earlier we used firewood. When it rained, we couldn’t cook. The kerosene stove was also difficult to use. Now it is much better.”(landless female, ST, 37 years)

On prompting whether fluoride levels in groundwater may have reduced because of groundwater recharge, a manager felt it may have happened “*to a small extent*”, mentioning “*this had been tested*” in another project site (NGO manager 2). Fluorosis was of concern locally, and participants were aware that “*fluoride has increased in groundwater*” (land-owning male, GC, 42 years). It was suggested that local people should consider “*surface water harvesting*” and increase “*groundwater recharge*” to address these challenges (senior WSD practitioner 1). A sample rainwater harvesting system including a “*sand filter was also installed*” at one farmer’s home, which he was still using (land-owning male, GC, 63 years). Finally, on enquiry about farm ponds as potential hazards for drowning (based on information from an informal conversation with a health worker), local people mentioned that this had not occurred in their villages, but that they were “*now required to fence the ponds*” (land-owning male, GC, 63 years).

## 4. Discussion

### 4.1. Size and Distribution of Impacts of WSD Projects

The impact of WSD projects perceived on agriculture, water availability and livelihood reported from remote parts of Maharashtra [23,24,26,27,28] were not uniformly seen in our study areas in Kolar district. This might, at least partly, be explained by established groundwater exploitation, piped water supply and small land-holdings prior to the development and operation of the WSD projects, coupled with close proximity to a large city (Bengaluru). However, sizable impacts were experienced by some large dry-land horticultural farmers and poor landless labourers. An earlier evaluation from Karnataka also showed that only few households benefitted substantially from WSD projects [14]. Literature also corroborated that farmers with large holdings showed stronger impact and were in a position to disproportionately exploit groundwater [5,45]. Moreover, use of water-conserving technologies such as drip irrigation is encouraged by WSD policy [12] for synergistic impacts with WSD structures, especially benefiting larger farms. However, recent WSD projects have been more equitable in design due to the inclusion of livelihood support for landless households [13]. Several small-holder farmers either did not sustain the structures or did not even participate in the projects due to perceived wastage of land, as was also found elsewhere [14]. It may be useful to study motivation behind participation in WSD projects. A relatively small proportion of SC and ST families owned land, which were reportedly of smaller size and lower quality compared to general category households, as corroborated by literature [46]. However, these households participated and benefitted similarly to what was experienced by families of other castes with comparable land-holdings. Impacts of WSD projects on water conservation and agricultural productivity have been well established in the literature [22,47].

Women provided greater detail on health impacts, as most of the relevant project activities leading to these impacts (e.g., livestock management, livelihood training and awareness sessions) were coordinated through SHGs, and also the related household chores (e.g., cooking, water collection and livestock rearing) were primarily managed by women. While SHGs have also been created and supported outside of WSD projects, and the benefits of SHGs have been established [12], WSD projects provided scope for identifying and engaging all households that would benefit from small grants, and hence, provide systematic ground-level support over a few years. WSD projects without a strong livelihood component may miss an opportunity to benefit the poorest households, especially those unable to travel for work.

### 4.2. Interpreting Nutritional Impacts

Experience from India suggests that access to irrigation, ownership of livestock and elevated income had positive impacts on household nutrition [48]. In our study, poor landless households that took up livestock management and vocational support reported improved access to nutritious food. The few small farmers who maintained WSD structures also reported enhanced food security through higher survival and production of staple crops during poor monsoons, as reported in an previous study from the region [12]. Earlier studies found a shift to commercial crops and reduction in cultivation of staples [12,23]. Reduction of staples was not seen in our study area. Indeed, food security remained uncompromised, and nutritional quality was enhanced, pointing to impacts that depend on the specific contexts. A study reported that WSD projects led to increased irrigation and consequently increased workload on women [23]. Women in our study area perceived increased workload after installing borewells or farm ponds, potentially resulting in a negative impact on nutritional status [48,49]. However, while increase in the number of borewells has been noted in WSD project areas, it is not clear whether this is actually due to WSD development or other factors [12], as borewells are not promoted in WSD. Additionally, while fish rearing has been encouraged in surface water bodies by WSD policy [13], local people faced challenges with the implementation.

### 4.3. Interpreting Other Health Impacts

Although local people and NGO officials did not report any increase in mosquito nuisance, the potential for mosquito-breeding in farm ponds, check dams and water troughs has been documented in earlier studies in the same region [50,51]. At present, malaria is not a concern locally, and Japanese encephalitis has not occurred in Kolar district for few decades. However, malaria cases have been reported from neighbouring districts [52].

Allied activities that were conducted as part of WSD projects such as medical camps and health insurance enrolment were utilised only by a few households. Hence, the strategy for and utility of such activities should be reviewed.

High fluoride levels in groundwater—a concern in Kolar district—can lead to dental and skeletal problems [53,54], and potentially affect neuro-development in children [55]. Rainwater harvesting and groundwater recharge were encouraged as part of WSD projects in the study area, with evidence suggesting reduced exposure to high fluoride levels [56,57]. There were also consistent efforts to help local people leverage governmental welfare schemes, which was possible because of the NGO’s commitment. This resulted in improving access to clean fuels, drip irrigation and sanitation, which contributed to improving people’s health and well-being. Entry-point activities, such as through constructing a water tank, created impact only in villages without basic services. These activities also contributed to stress reduction, alongside better livelihood management. These kinds of activities and impacts cannot be expected consistently across WSD projects in India as they are context dependent.

Diarrhoeal diseases were not a concern in the study area, in contrast to findings from remote villages in Maharashtra, where WSD projects contributed significantly to sanitation practices and outcomes [26,27,28]. Most study villages had improved access to water and toilets prior to WSD projects. Similarly, while other studies indicated reduced migration following the implementation of WSD projects [12,27], in our study only a few of the households that adopted livelihood opportunities showed reduced needs to travel for livelihood. This finding too was related to contextual factors. Drowning in farm ponds were reported from other parts of the district [58], and people in the study area were aware of this risk and the need to install fences.

### 4.4. Challenges in Eliciting Health Impacts

Health impacts of WSD projects, whether positive or negative, were largely incidental. Perceived health impacts were often elicited on probing, as was also noted by Pandit [23]. NGO officials were able to articulate more about health and other impacts possibly because they: (i) were more familiar with the theoretical basis for project activities and their anticipated impacts; (ii) were better aware about conducted activities; and (iii) knew about benefits experienced across villages and projects. Their accounts provided insights into both the efficacy (i.e., outcome expected in ideal circumstances) and effectiveness (i.e., outcome observed in real world situation) of interventions. Local people were limited to the experience of their own households and local area, besides complexity in interpreting the contribution of factors such as weather fluctuation and other governmental programmes. The size and distribution of impacts on agriculture and livelihood had implications for health impacts, which in turn are dependent on project quality and context.

### 4.5. Strengths of the Study

The NGO that planned and implemented these WSD projects was experienced, which is an important factor explaining the overall high quality of the projects. The WSD projects examined in this study had been completed between 2 and 10 years prior to the interviews, which helped understand how activities and impacts were appreciated and sustained. A range of participants were interviewed to better understand how and why insights may wary between actors. Interviews were preferred to FGDs because of the need to probe individual and household-level experiences and impacts. Pictures taken of WSD structures were used in interviews with the academician to help contextualise the discussion. These features provided a strong and rounded impression of perceptions of impacts, opportunities and risks of WSD projects on health. Literature on adaptation strategies in semi-arid regions that are vulnerable to climate change is emerging [59]. Our study contributes to this through a health lens, which has often been lacking.

### 4.6. Limitations, and Scope for Further Research

As the study attempted to deepen the understanding of the full range of health and health-determining impacts of WSD projects, some aspects may not have received the required depth of attention as may have been possible if the scope was narrower. Additionally, interviews and coding were done by one person. There is scope for mixed method studies (i.e., combining quantitative and qualitative approaches), especially on nutritional impact, vector-breeding in water-holding structures, zoonotic diseases through livestock management and water quality. Other aspects that should be explored further are migration and mental health impacts, which are relevant to drought-prone agricultural communities [60]. Using a resilience framework [61] or a health access livelihood framework [62] may help plan and study projects such as WSD. Quantitative studies through the use of appropriate methods such as structural equation models would further contribute to the understanding of the size of impacts of WSD projects, due to the presence of multiple impact pathways and multifactorial outcomes.

### 4.7. Future Action

We planned this study to inform an HIA of a newly planned WSD project in Kolar district. HIA are applied prospectively to identify and judge potential health impacts of planned projects, programmes or policies and recommend measures to minimise negative and maximise positive impacts [63]. Through the current study, we identified a range of positive and potentially negative health impacts of WSD projects. This list of health impacts was used to identify gaps in health data for the project villages of the planned WSD project at baseline. The data gaps were then addressed through a baseline survey conducted in the project villages and comparison villages, which can then be monitored and later compared with baseline health status after project completion. This would give quantitative insights on the size of impacts of WSD projects and the utility of the HIA. These actions would demonstrate approaches to address critical cross-sectoral health policy concerns identified by the Government of India such as undernutrition and sanitation [64], and help realise the vision of WSD projects [13,30].

## 5. Conclusions

Our study revealed that WSD projects in semi-arid settings of southern India were perceived to impact various health outcomes and determinants. The identified pathways to health impacts were often secondary to the primary impacts on agriculture, livelihood and local ecology. We also found that WSD projects offered a platform for linking local people more effectively to existing health-related and health-determining schemes, especially because of involvement of all households through the SHGs. Health-wise, those benefitting most may be the poor landless households that were able to effectively utilise and build upon the livelihood support. These families experienced better nutrition through mechanisms such as increased milk production in the household and higher income. However, potential for vector-breeding increased because of increase in water-holding structures, though local people and NGO workers did not perceive this as a risk. In addition, through comparison of findings with existing literature, it was demonstrated that impacts for such WSD projects are context-dependent—related to local geography, socioeconomic factors and developmental status. There is scope for integrating health activities and monitoring more systematically into these projects, especially for nutrition and VBDs. This would be useful as they are key health policy concerns for rural India, and also in semi-arid areas in other low- and middle-income countries. We have attempted this integration through an HIA of a proposed WSD project locally, which will be published as a case study separately.

## Figures and Tables

**Figure 1 ijerph-17-03448-f001:**
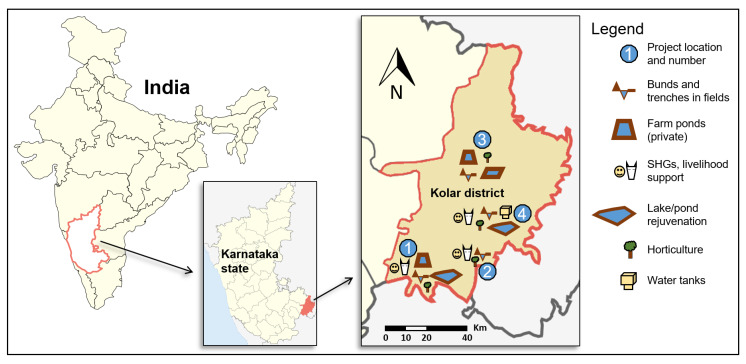
Map showing approximate locations of completed WSD projects included in this study and components of each project (modified from images by *3xK* and *PlaneMad*, respectively, distributed under a CC-BY 2.0 license). SHG, self-help group

**Figure 2 ijerph-17-03448-f002:**
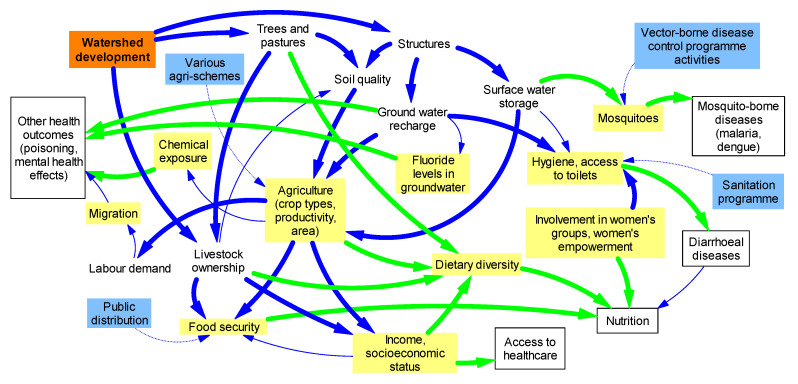
Potential pathways to health impacts of WSD projects (thick blue arrows indicate project activities; blue boxes and dotted arrows indicate governmental programmes; yellow boxes indicate health-determining impacts; thick green arrows indicate potential health impact pathways; white boxes indicate potential health impacts of WSD projects).

**Figure 3 ijerph-17-03448-f003:**
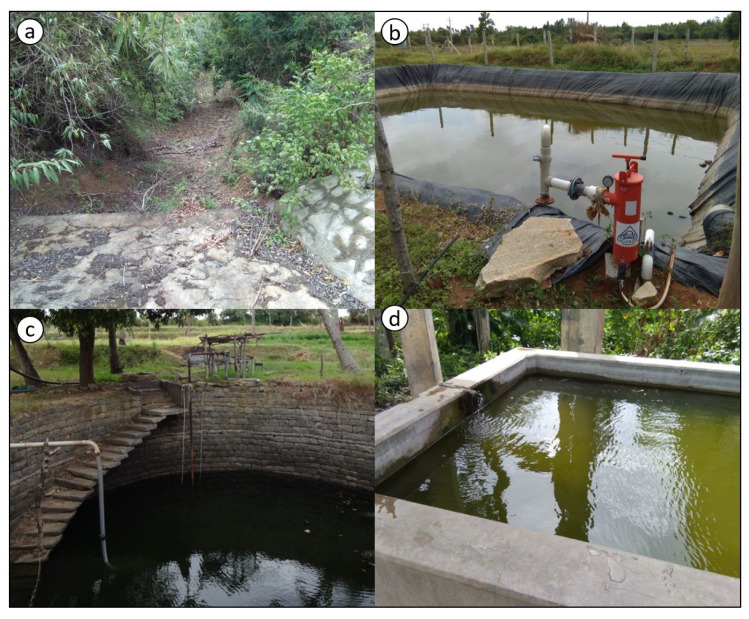
Water-holding structures created and encouraged as part of WSD projects: (**a**) check dam (without water); (**b**) farm pond (with tarpaulin lining); (**c**) open well; and (**d**) water trough for cattle (source: first author; images captured during transect walks in study villages).

**Table 1 ijerph-17-03448-t001:** Characteristics of the four study villages in Kolar district, southern India where watershed development projects were implemented (taken from [34]).

Village Characteristics	Village 1	Village 2	Village 3	Village 4
Sub-district	Malur	Bangarpet	Kolar	Bangarpet
Distance to sub-district headquarters (km)	26	44	10	20
Total households	181	187	92	46
Total population	879	676	454	211
Scheduled caste population (%)	15.6	47.0	15.4	0.0
Scheduled tribe population (%)	24.6	6.5	0.0	100.0
Geographical area of village (ha)	440.6	777.6	87.2	151.8
Distance to primary health centre (km)	5–10	5–10	5–10	5–10
Water supply	Available	Available	Available	Available
Governmental crèche (*anganwadi*)	Available	Available	Available	At nearby village

**Table 2 ijerph-17-03448-t002:** List of codes and themes used during the analysis.

Themes	Sub-Themes	Codes
Pathways towards nutritional impacts	Direct impacts on food security and nutrition; income-mediated impacts on food security and nutrition	Crop choices, irrigated agriculture, rain-fed agriculture, financing agriculture, agri-inputs, wildlife and pests, surface water irrigation, food security, diet, wage labour, livestock, self-help groups, common lands, migration, nutrition, chemical toxicity, participation in WSD project, tree-planting, livelihood activities, impact on water and soil conservation, impact on food production, impact on wage labour, impact on livelihood and impact on nutrition
Impacts on disease vector ecology		Mosquitoes, vector-borne diseases, watershed structures, impact on local environment, impact on mosquitoes and vector control activities
Health impacts of opportunistic activities		Hygiene, drinking water quality, diarrhoeal diseases, other work-related health problems, access to healthcare, creating local institutions, other structures created, health activities, water for domestic use, impact on awareness, impact on sanitation and other health impacts

**Table 3 ijerph-17-03448-t003:** Details of interviewed persons.

Participants	Number (%)	Average Duration - Minutes (SD)
Local people (*n* = 26)		
Men [average age (SD): 40.5 years (11.4 years)]	13 (50)	23.4 (10.1)
Women [average age (SD): 46.2 years (12.7 years)]	13 (50)	19.6 (7.9)
Scheduled caste	4 (15)	15.5 (2.4)
Scheduled tribe	9 (35)	16.8 (6.4)
Other caste	13 (50)	26.6 (9.4)
Land-owning, with irrigation	12 (46)	25.4 (7.9)
Land-owning, without irrigation	5 (19)	21.5 (15.7)
Landless	9 (35)	19 (2.5)
Key informants (*n* = 14)		
Field health-worker	5 (35.7)	16.8 (4.6)
Field project staff (liaison)	4 (28.6)	29 (6.2)
Project managerial officials	3 (21.4)	37.3 (18.5)
WSD expert (other region)	1 (7.1)	92
Environmental health academician	1 (7.1)	81
FGD participants (*n* = 2; with 12 participants)		
Female [average age (SD): 42.5 years (14.5 years)]	12 (100)	27 (5.7)

FGD, focus group discussion; SD, standard deviation; WSD, watershed development.

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
