# Peer review of "Perceived Health Impacts of Watershed Development Projects in Southern India: A Qualitative Study"

_ijerph, 2020, doi:10.3390/ijerph17103448_

Round 1

Reviewer 1 Report

Watershed Management bring about the best possible balance in the environment between natural resources on the one side and man and animals on the other. WSD projects have considerable potential for community health impact through influencing agriculture, water and livelihoods. The manuscript “Perceived health impacts of watershed development projects in southern India: a qualitative study” (ID: ijerph-795792) revealed that WSD projects in semi-arid settings of southern India were perceived to impact various health outcomes and determinants. The selected topic has high theoretical significance. This research strengthens the practical understanding of health impact of WSD projects. However, there are some scientific problems in the current status of this manuscript. The detailed comments and suggestions are listed as follows:

1) Line 12: Regarding the abstract, it can be shortened and more concise and more prominent research purpose and significance; explain the research background.

2) Line 30: Too many keywords, should be control the keywords number of 3-5.

3) Line 94: The article lacks the evaluation of the significance and value of this research.

4) Line 113: We believe that the author should give the number of people in each village in table 1 to further compare with the actual number of people interviewed.

5) Line 117: Why 40 people were selected as the research object, whether it conforms to the characteristics of the local population, and what sampling method was adopted.

6) Line 124: Why not at the same time. The cause of the two surveys?

7) Line 143: These principles of content selection.

8) Line 97: Regarding the Materials and Methods, The explanation of the theoretical method is not detailed enough in the study. In my opinion, the data sample size is not enough. The data should be better collated and analyzed in more detail and at a deeper level.

9) Line 199: The author lacks the representativeness and validity analysis of the number of respondents.

10) Line 202: The author only conducted a descriptive study on this part of the content. We believe that this kind of study lacks quantitative consideration and may produce better results through semi-structured questionnaire survey and quantitative analysis.

11) Line 389: The author should give hints of future action. It seems to me that we can synthesize a paragraph (4.4, 4.5 and 4.6).

12) Line 489: In fact, in related research fields, many quantitative methods are adopted, such as structural equation analysis based on actual measurement data.

In summary, the data of this manuscript is insufficient, and the research method is difficult to quantify and reveal the reasons for the influence. I recommend major revision of the manuscript, and asking for additional data before reviewing the manuscript, and hope of further quantifying the health impact of the WSD based on additional data.

Reviewer 2 Report

The manuscript entitled “Perceived health impacts of watershed development projects in southern India: A qualitative study” by Adithya Pradyumna, Arima Mishr, Jürg Utzinger and Mirko S. Winkler is carefully organized and well written. The only potential weakness that I see is the relatively small sample size for making generalizations about community perceptions. But given the qualitative nature of the study it’s not clear how or if a larger pool of individuals would have influenced the interpretations and findings of the authors.

My only suggestion for improvement would be to be more specific about the findings in the Abstract. I realize that this is an exploratory study to be followed by a case study later, but the language in the Abstract should allow a reader to take away some specific information on what the findings of this study are. Currently it mentions that wide range of factors “were reportedly impacted to varying degrees” and that some of the findings “have not been previously reported” but a reader will have to read the entire document to find out any specifics beyond these vague statements. So, in the last sentence where the authors state that their “findings may encourage…organizations to incorporate health considerations…” the reader still has little to no idea why.

One minor correction to Table 1: The unit (presumably km) was left out of the line “Distance to sub-district headquarters.”

Round 2

Reviewer 1 Report

Watershed Management bring about the best possible balance in the environment between natural resources on the one side and man and animals on the other. WSD projects have considerable potential for community health impact through influencing agriculture, water and livelihoods. The manuscript “Perceived health impacts of watershed development projects in southern India: a qualitative study” (ID: ijerph-795792) revealed that WSD projects in semi-arid settings of southern India were perceived to impact various health outcomes and determinants. The selected topic has high theoretical significance. This research strengthens the practical understanding of health impact of WSD projects. The author has revised the three contents of the five questions we raised. In the end, consider the research significance and academic value, I recommend receive the manuscript.